# Effect of Al Layer Thickness on the Bonding and Mechanical Behavior of a Mg-(Al-)Ti Laminated Sheet Prepared by Hot-Rolling after Differential Preheating Treatment

**DOI:** 10.3390/ma15082805

**Published:** 2022-04-11

**Authors:** Wenbo Luo, Yunzhe Feng, Zhiyong Xue, Qinke Kong, Xiuzhu Han

**Affiliations:** 1Institute for Advanced Materials, North China Electric Power University, Beijing 102206, China; luowenbo@ncepu.edu.cn (W.L.); fengyunzhe@hotmail.com (Y.F.); kongqinke@hotmail.com (Q.K.); 2Beijing Institute of Spacecraft System Engineering, Beijing 100094, China

**Keywords:** Mg-Ti alloy, hot-rolling, bonding interfaces, mechanical property, composite sheets

## Abstract

Mg-(Al-)Ti laminated sheets with large bonding interfaces were prepared by a differential temperature hot-rolling process, in which the preheating treatment of Ti was 25–100 °C higher than that of Mg. The rolled sheets contained different Al layer thicknesses (≤0.05 mm), and the thickness of the diffused region at the interface of 3–7 μm was formed by rolling at 175 °C. The interfaces were the solid-solution regions of Mg(Al) and Ti(Al), and no intermetallic compounds were generated during both the rolling process and annealing treatment. The hardness of the interfaces was 16–30% greater than that of the Mg matrix and Ti matrix. The results of mechanical tests displayed that the Mg-(Al-)Ti sheets exhibited higher strength and elastic modulus compared to those of the rolled AZ31B sheet. Their UTS and YTS were about 223–460 MPa and 303–442 MPa, respectively, with an elongation of 0.04–0.17 and high elastic modulus of 52–68 GPa. The Mg-Ti (containing about 62 at.% Mg) rolled sheet exhibited the most excellent strength. The UTS and YTS were about 460 MPa and 442 MPa, with an elongation of 0.04 and elastic modulus of 61.5 GPa. Additionally, Mg-Ti sheets with thin Ti thickness possessed a higher work-hardening rate (*n*), as well as hardening rate, than the rolled Mg-Al-Ti sheets. This is because fractured Ti pieces around the interfaces have a significant strengthening effect. This study provides a simple method for fabricating Mg-(Al-)Ti sheets with high elastic modulus.

## 1. Introduction

With the development of modern industries such as transportation and aerospace, it is difficult for conventional metals to achieve comprehensive performance in the mechanical and functional aspects of state-of-the-art applications. This drives the pursuit of advanced metals such as laminated metal composites (LMCs) consisting of dissimilar alloys, especially in terms of low manufacturing costs. Magnesium alloys have been used as the lightest metals due to their high specific strength and excellent damping capacity [1]; however, their poor ductility, low strength, and relatively small elastic modulus (E) restrict their wide application.

It is well known that the specific stiffness of Mg alloys is close to that of Al alloys and steel, while their absolute elastic modulus is still low, only about 44 GPa, which is much less than that of Al alloys and high-strength steel [2,3]. Regulating endogenetic high-modulus second phases or particles is an effective method to enhance the elastic modulus of Mg alloys, where the values of E can reach up to 49–53 GP [4,5,6,7,8]. For example, (RE-Si)-rich particles were added to Mg-Gd-Y-Zn-Mn alloy, and the E value reached approximately 49 GPa [7]. MgAg and Gd**_5_**Ge**_3_** phases were added to Mg-10Gd-1.5Ag-0.2Mn-3.5Ge alloy, and the modulus increased to about 53 GPa [8]. In addition, foreign high-modulus phases, such as SiC, Y**_2_**O**_3_**, SiO**_2_**, graphene, and even carbon nanotube (CNTs), were added to Mg alloy to form reinforced Mg matrix composites, as CNTs have perfect elastic modulus and strength, excellent thermal conductivity, and good electrical properties, and their elastic modulus reached up to about 200–950 Gpa [9,10,11,12,13,14,15]. For example, when 0.5–5 wt.% CNTs were added to AZ91D alloy, the elastic modulus increased from about 40 Gpa to about 43–51 GPa. Both the strength and elongation were significantly improved [14], while the elongation dramatically decreased with an increase in CNT content. However, the dispersibility of CNTs in the Mg matrix is a non-negligible challenge for industrial applications. Adopting Ni nanoparticles coated with graphene nanosheet (GNS) is an interesting strengthening method [13], but the interfacial bonding between Mg matrix and exogenous phases is still difficult to control [16,17].

Mg-based LMCs are usually prepared by severe plastic deformation (SPD), providing a simple method to prepare Mg alloys of high properties with high elastic moduli [13,14,15,16,17,18]. Different types of LMCs have been reported, including Mg-Al, Mg-Ti, Mg-Fe and other metals with high melting temperature [19,20,21,22]. For example, Liang et al., successfully prepared AZ31-5052Al and Ti-Al-Mg-Al-Ti sheets by adopting the hot-rolling process [20,23,24]. Ma et al., used the friction stir welding (FSW) process to obtain Mg-Al-Fe bonded sheets. It is well known that titanium (Ti) is an excellent lightweight metal with extraordinary mechanical properties and an elastic modulus that can reach up to 100 GPa. When Ti is added to Mg alloy, it can significantly enhance the high elastic modulus of the Mg matrix. However, when adopting metallurgic alloying methods, it is difficult to achieve direct bonding between Mg and Ti, as the melting point of Ti is approximately 1680 °C, which is significantly higher than the boiling point of Mg, which is 1090 °C. In addition, the solid solubility of Ti in the Mg matrix is nearly zero, and Mg alloys contain Ti in limited solid solutions or chemical reactions, resulting in no formation of Mg-Ti IMCs. Although some methods have been investigated to achieve good bonding between Mg and Ti, such as adopting the mechanical alloying method, bulk and stable nanocrystalline Mg-1.5 at.% Ti alloy was successfully prepared with ultimate compression stress of only 202 MPa [1]. When adopting the mechanical milling method, a very high Ti content of 3.18 at.% in the Mg matrix was obtained [25]. In contrast, the mechanical alloying process makes it difficult to obtain bulk, even large-sized, materials. Furthermore, Mg-Ti intermetallic phases were prepared using the severe plastic deformation method. Mg-Ti alloy contains four metastable Mg-Ti phases designed by high-pressure torsion (HPT) [3,26]. In addition to rolling AZ31/6061Al/TC4 metals, which exhibit high mechanical properties, the UTS and YTS are approximately 420 MPa and 380 MPa, respectively, with an elongation of 10% [27].

Owing to its large deformation resistance gap, an Al layer is usually used to support the bonding between Mg and Ti; however, Mg-Al IMCs can easily form around the bimetallic Mg-Al interface [25,28,29,30]. The effect of both the preheating treatment temperature and interdiffusion Al layer on the bonding interfaces and mechanical properties is unclear. In this work, we used Al layers of different thicknesses and adopted the rolling process with differential preheating treatment temperature to study their effect on bonding. We then successfully prepared the rolled Mg-(Al-)Ti sheets with interdiffusion interfaces. The bonding interfaces and mechanical properties were systematically studied. This study will provide a simple method to achieve the bonding of dissimilar alloys and even to obtain the heterogeneous structure of large-sized alloys for multifunction applications.

## 2. Materials and Methods

The raw materials of the laminated sheets were 1060 Al (0.2, 0.05, 0.01 mm, 100 mm × 200 mm), AZ31 (1.3 × 100 × 200 mm), and TA1 (0.01, 1 mm, 100 × 200 mm), and their chemical compositions are listed in Table 1 and Table 2. The AZ31B sheets were first annealed at 180 °C for 4 h. The TA1 sheets were then annealed at 500 °C for 4 h. The surfaces of these sheets were brushed using a grinding machine and degreased using ethyl alcohol.

Before the hot-rolling process, the AZ31B sheets were heat treated for about 30 min at a temperature of 200 °C, and the TA1 sheets were kept for 30 min at 225, 250 and 300 °C (preheating treatment). After that, the three sheets were stacked layer by layer, in a sequence of Mg/Al/Ti/Al/Mg (5 layers) with a thickness of 3.62–4.00 mm, Mg/Al/Ti/ Mg (4 layers) with a thickness of 2.62–3.61 mm, or Mg/Ti/Mg (3 layers) with a thickness of 2.61–3.60 mm, as shown in Figure 1. The stacked sheets were subjected to hot-rolling for 3 passes at RT and 175 °C, with a rolling rate of 0.17 m/s, resulting in a total thickness reduction of about 75%. The thickness reduction of the first-pass was about 45%. The rolled samples are listed in detail in Table 3. Then, the stress-relief annealing of the rolled sheets was conducted at 200 °C, kept for 4 h, and cooled in air. Figure 1 shows a schematic diagram of the rolling and heat treatment process.

The samples for microstructural investigation were cut to the dimensions of 8 mm × 10 mm in the TD plane (cross-section, along the transversal direction, TD). They were then mechanically polished to a mirror-like surface using abrasive paper and diamond polishing paste. The microstructure was characterized by the Tescan Mira 3 field emission scanning electron microscope (SEM) equipped with an Oxford Instruments EDS system (Oxford Instruments, Oxford, Britain) and electron backscatter diffraction system. The phases were identified using X-ray diffraction (XRD) (Rigaku Ultima IV 3 KW, Rigaku Corporation, Tokyo, Japan, Cu-Kα radiation, at 40 kV and 300 mA) with 2θ ranging from 20° to 80° at a scanning rate of 0.02° s^−1^. The samples for the XRD analysis were from the normal and transversal direction planes (ND × TD). To explore the effect of rolling reduction on mechanical properties, dog-bone-style tensile specimens were cut along the rolling direction (RD) with a gauge length of 60 mm.

Tensile tests were performed using an MTS Criterion42 (MTS Systems, Eden Prairie, MN, USA) equipped with an extensometer. The strain rate was about 10^−3^ s−^1^, and each test was repeated three times at room temperature. The tests were based on the ASTM-B557-2015 standard, and the elastic modulus tests and analyses were based on ASTM-E111-2004(2010). Nanoindentation tests were performed using a Hysitron TI 980 TriboIndenter manufactured by Bruker (Bruker Corporation, Billerica, MA, USA) with a high load resolution (50 nN) and a high displacement resolution (0.01 nm). The hardness and Young’s elastic modulus were recorded to determine the bonding of the Mg-Al and Al-Ti interfaces. The measurements parameters were as follows: maximum load P_max_ = 3000 μN, and the depth in the range of 100–1000 nm under different zones: ~300 points (containing about 120 points for the matrix, at least 2 times for different places, and 30 points for line scanning around the interfaces each test, at least 2 times for different places) were impressed and analyzed around the Mg-Al and Al-Ti interfaces.

## 3. Results and Discussion

### 3.1. Interface Morphology of the Rolled Mg-(Al-)Ti Sheets

Figure 2 shows the X-ray diffraction (XRD) patterns of the rolled Mg-Al-Ti sheets at high temperature. The thickness of Al was 0.05 mm, 0.01 mm and 0 mm. It contained three phases: α-Mg, Al, and α-Ti and no IMCs observed. The main peaks of these phases were (10–11)_Mg_, (10–11)_Ti_ and (111)_Al_.

Figure 3 shows the SEM morphology of the interfaces of Mg-Al and Al-Ti at different temperatures (RT, 300 °C) in the rolled Mg-(Al-)Ti sheets. No apparent voids, discernible defects, and IMCs were observed in the interfacial regions. Figure 3a,b shows the microstructure of the interfaces at RT. Owing to the severe plastic deformation effect, a wavy surface was formed on the Ti side, which indicates good bonding of the Ti layer with the Al layer; however, the sheets encountered a macroscopical fracture along the transversal direction (TD), as shown in the inset of Figure 3a. Upon increasing the preheating temperature of the Ti sheet (before the rolling process), the deformation of Ti was more severe with an increased number of serrated zones around the Al-Ti interfaces. The wavy serrated interfaces indicated the fresh metal (Ti) would squeeze into bonding with the opposite metal (Al), as shown in Figure 3c–h. The reduction of the Al layer was about 80%, and the significant rolling reduction decreased the middle plastic asymmetry during the rolling process [29] to assist the bonding between different metals.

Table 4 shows the EDS results for the Mg-Al and Al-Ti interfaces shown in Figure 3, which contained about 4.2–8.6 at.% (Al + Ti) around the Mg side (Mg matrix) and about 4.8–9.5 at.% (Al + Mg) in the Ti matrix around the Al-Ti interface. This indicates that it encountered interdiffusion in the interfacial regions, thereby forming a Mg and Ti matrix solid solution.

To investigate the diffusion behavior around the interfaces, EDS line scanning analysis was conducted to determine the diffusion thickness, as shown in Figure 4. The widths of Mg-Al and Ti-Al were approximately 3 μm and 2.5 μm (rolling at RT), respectively. With an increase in temperature, the interface thickness of Mg-Al increased to 5.2, 5.5 and 5 μm. The thickness of Ti-Al increased slightly to 3 μm at 225 °C, and it did not change with an increase in the Ti sheet temperature. This is because the high temperature promoted the diffusion of Al and had little effect on Ti. When the heating temperature was increased to 300 °C, the surface of Ti alloy was slightly oxidized, hindering the interdiffusion of Ti and Al atoms. A similar diffusion layer was observed in the Mg-Al bonded joint with a Ni layer, which was welded at 430–440 °C for 90 min; its diffusion thickness was in the range of 2.2–10.6 μm [30]. Generally, the width of the diffusion interface changed slightly with an increase in the preheating treatment in this study. This is because the temperature (including rolling temperature) had a slight effect on the atom interdiffusion, and the diffusion occurred for a short time in the rolling process. This could be verified in other Mg-Al-Ti rolling sheets; for example, the Ti/Al/Mg laminated composites, in which the interdiffusion thicknesses, including the Mg-Al and Al-Ti interfaces, rarely changed, even with rolling at higher temperature, i.e., 400 °C [31].

Figure 5 shows the microstructure and EDS mapping for different Al layer thicknesses (0.05 mm, 0.01 mm and 0 mm). It can be seen that the diffusion of the Al atom into the Mg side was more significant than the Ti side with decreasing Al layer thickness. The Mg-Ti interface also encountered diffusion without the Al layer. The corresponding EDS line analysis results are shown in Figure 5j-l. The width of the Mg-Al interface was about 5.2 μm in the Mg-0.05 mm Al-Ti (0.05Al) sheet, and it increased slightly in the Mg-0.01 mm Al-Ti sheet (0.01Al) by about 6 μm. Meanwhile, for the Ti-Al interfaces, the width was approximately 3 μm in both the 0.05Al sheet and the 0.01Al sheet. This implies that the decreased Al layer has a slight effect on atom diffusion around the Mg-Al and Al-Ti interfacial regions. For the Mg-Ti sheets without the Al layer, there is only one type of interface, Mg-Ti, which forms the interdiffusion of Mg-Ti interfacial regions with a width of 4 μm.

Based on the solid-state diffusion theory of metals, the following factors are necessary for the diffusion of different metal atoms: the potential chemical gradient, solid solution, temperature, and diffusion time. It is well known that there is rarely a negligible solid solubility of Ti in Mg alloys. For the Mg-Al-Ti sheets, the Al layer with interdiffusion was the key for the interfacial bonding, especially for the Mg-Al and Al-Ti interfaces; Al could dissolve into both the Mg and Ti matrices, which is the driving force for the formation of the transitional interfacial region. Additionally, the deformational temperature has a vital effect on the interface between the Mg and Ti alloys. The higher preheating temperature of the Ti alloy weakens its deformational resistance during the rolling process, and the deformed resistances of the different layers of metals are significantly reduced. Fresh metal was then squeezed out of the oxide layer and contacted the interfaces [31]. With rolling for multiple passes, interfacial microvoids were gradually formed, in which the thin oxide fragments were scattered, the exposed fresh metals constantly contacted, and the atomic bonding interface gradually formed.

### 3.2. Mechanical Behavior of the Rolled Mg-(Al-)Ti Sheets

#### 3.2.1. Nanoindentation Tests on the Mg-Al and Al-Ti Bonding Interfaces

Nanoindentation was used to determine the interface bonding. The indentation matrix was about 25 μm × 60 μm around the Mg-(Al-)Ti interfaces containing the Mg, Al and Ti matrix, containing about 120 nanoindentation points each test, and 2 tests were conducted in this study, the whole test data was povided in Appendix A and Appendix A (Appendix A). Figure 6a shows the hardness in the Mg side was about 0.6 ± 0.3 GPa, and the hardness significantly increased to 0.8 ± 0.3 GPa in the Mg-Al interfaces and 2.5 ± 0.3 GPa in the Al-Ti interface. Similarly, the elastic modulus gradually increased from about 45 GPa in the Mg matrix to about 69 GPa at the Mg-Al interface and 100 ± 10 GPa around the Al-Ti interface, as shown in Figure 6b. It is well known that the elastic modulus is not structurally sensitive like strength [31,32,33]. The gradual change of both hardness and elastic modulus around the two interfaces shows the bonding of dissimilar metals; therefore, the significant bonding of the interfaces between Al, Ti and Mg is vital for the rolled Mg-(Al-)Ti sheets.

The nanoindentation continued to conduct around the Mg-Al and Al-Ti interfaces, and both the hardness and elastic modulus results are shown in Figure 6c,d. At the Mg-Al and Al-Ti interfaces, the hardnesses were about 1.3 GPa and 4 GPa, which were about 30% higher than those of the Ti matrix and the Mg matrix, i.e., 1 GPa and 3 GPa. Additionally, the values were about 140 GPa and 60 GPa at the Al-Ti interface and Mg-Al interface, respectively, as shown in Figure 6d, which were significantly greater than those of the elastic modulus of the Ti matrix and the Mg matrix, i.e., about 120 GPa and 50 GPa.

#### 3.2.2. Tensile Mechanical Behavior of the Rolled Mg-(Al-)Ti Sheets

Figure 7a shows the true tensile stress–strain curves of the as-rolled Mg-(Al-)Ti sheets. The ultimate tensile strength (UTS), yielding tensile strength (YTS), and elongation to break (EL) values are also listed in Table 5. The thickness of the Mg-(Al-)Ti sheets with different Al layers had a certain effect on the stress and strain. A thicker Al layer positively promotes elongation at the expense of stress. With Al layer thickness decreasing from 0.05 mm, 0.01 mm, to 0 mm, the values of UTS were about 399, 387 and 424 MPa, and the corresponding YTS values were 358, 354 and 408 MPa, respectively. The EL values were reduced from 0.05 to 0.03 and 0.01. For the Mg-Ti sheet, especially the Mg-0.01Ti-Mg (0.01Ti) and Mg-0.01Ti-0.01Al-Mg (0.01Ti-0.01Al) sheets, the strengths were relatively small, approximately 329–349 MPa.

The preheating temperature of Ti has a significant effect on its mechanical properties. When the temperature increased from 225 °C to 250 °C, the strength of both increased significantly. The UTS was 399 and 460 MPa, and the YTS was about 358 and 442 MPa, respectively, with elongation of 0.05 and 0.04. As the temperature increased to 300 °C, both the strength and elongation decreased, and the UTS and YTS were about 424 MPa and 401 MPa, with a low elongation of 0.01.

Figure 7b shows the mechanical curves of the rolled sheets after annealing at 200 °C. The corresponding values of the UTS, YTS, and EI are listed in Table 6. The strengths of the annealing rolled sheets were all reduced, and at the same time, the elongation increased significantly. The UTS slightly decreased by about 1–14% (3–61 MPa) to 303–418 MPa, while the decrease in UTS was very small for the 0.01Al and 0Al sheets. The 0.05Al sheets (Mg-0.05Al-Ti) had a relatively large UTS drop no matter the heat treatment of raw sheets (about 10%). All the YTS exhibited a significant reduction after the heat treatment, decreasing by about 23–35% to the range of 223–339 MPa. The elongation of all the annealing samples increased substantially by more than 80%. Their values were located in the range of 0.09–0.18, especially for the 0.01Ti (Mg-0.01Ti) and 0.01Ti-0.01Al sheets, whose elongations were significantly increased.

Higher UTS and elongation, 373–408 MPa and 0.09–0.15, were achieved in the Mg-0.01Al-Ti (0.01Al) and Mg-1Ti (0Al) sheets than that of the Mg-0.05Al-Ti (0.05Al) sheet with a thicker Al layer of 363 MPa and 0.09, indicating suitable bonding interfaces between Mg and Ti/Al. Both strength and elongation decreased after the preheating treatment at 300 °C before the rolling process. This is mainly because of the formation of the oxide layer at higher temperature, and the bonding of the Al-Ti interfaces is relatively weak. Compared to the other reported Mg-based dissimilar sheets, Mg-steel and Ti-Al-Mg, rolled Mg-(Al-)Ti sheets show excellent compressive mechanical properties in this work under the high Mg content condition (>60 at.%). The reported Mg-based LMCs were mainly Mg-steel, Mg-Al-TC4 and Mg-Al-TA2, and their strengths and elongations show large differences. The mechanical properties were higher with a decrease in Mg content, as indicated in Table 6. The TYS of the studied Mg-Al-Ti rolling sheets (250Ti) is significantly higher 50 MPa than that of Mg-steel and Mg-Al-TA2 alloy, and the elongation is also larger than that Mg-steel. The compressive mechanical properties were similar to those of the Mg-Al-TC4 rolling sheets. The UTS and TYS were about 459 MPa and 380 MPa, with an elongation of 0.08; however, the Mg content of the latter alloy was so low, about 22 at.%, which is not a Mg-based alloy.

The elastic modulus of these alloys is also analyzed in this section. High elastic modulus was successfully achieved by bonding Mg with Ti and Al alloys. A high E was achieved in the rolled Mg-0.01Ti, Mg-0.01Al-0.01Ti and Mg-1Ti sheets; its values were approximately 50–56 GPa, which were higher than those of Mg alloys containing a high content of RE elements [35,36]. A higher E was obtained for the 0.01Al-1Ti and 0.05Al sheets, about 62–68 GPa. The elastic modulus of the rolled sheets was slightly higher than those of the calculated values based on a mixture of rules, mainly because the rolled sheets formed highly diffusing bonding interfaces, and the interfaces showed superior mechanical properties compared to the Mg matrix and Ti matrix.

Figure 8a shows the fracture morphologies of the Mg-(Al-)Ti rolled sheet; the fracture is approximately flat, rather than the AZ31-Al-TC4 rolled sheet, in which the fracture is a classic shear feature with an angle of 45°. Cracks between Mg, Ti, and Al were observed in the rolled sheets. Many dimples were also found on the fracture surfaces of AZ31B and the Ti sheets, and the depths of these dimples on the Ti sheets were more significant than those on the AZ31B sheets, as shown in Figure 8b.

The fracture morphologies of the Mg-Ti sheets without Al layers were also investigated, as shown in Figure 8b. The two interfaces of the Mg and Ti alloys also completely cracked. Many dimples were also found on both the Mg and Ti surfaces, while the size of the dimples on the Ti surfaces was slightly larger than those on the Mg surfaces, which indicates better flexibility in the plastic deformation. In addition, Ti alloy exhibited significant necking after the heat treatment. Some strong bonding points were found around the interfaces between Mg and Ti alloys, which indicates that more stability and bonding were achieved by conducting heat treatment at a lower temperature.

#### 3.2.3. Hardening Effect of the Rolled Mg-(Al-)Ti Sheets

Figure 9 shows the strain-hardening rate of the rolled sheets used to determine the strengthening effect during the plastic deformation stage. The strain-hardening rate shows typical decreasing hardening with an increase in strain, and its values are mostly in the range of 500–2000 MPa. The annealed Mg-(Al-)Ti sheets showed a slightly higher strain-hardening rate than the others. The differential preheating temperature had a significant effect on the strain-hardening rate of the 300Ti and 250Ti samples, showing a higher strain-hardening rate at the beginning of the plastic deformation stage than that of 200Ti. Interestingly, the Mg-Ti sheets containing Mg-1Ti (0Al) and Mg-0.01Ti (0.01Ti) showed a higher hardening effect than rolled sheets containing Al layers. Additionally, the Mg-0.01Ti-0.01Al rolled sheets showed distinct hardening behaviors; their hardening rate had a significantly larger increment and decrement, as indicated in Figure 9b. This indicates that the dissimilar metals, Al and Ti, enhanced the Mg matrix.

During the deformation process, they abruptly crack down with an increase in strain. Figure 10 shows the SEM images of the 0.01Ti and 0.01Ti-0.01Al sheets, and it can be seen that the interlayers fractured to pieces with a width of about 2–6 μm. The deformational discontinuous Ti pieces were still regarded as the strengthening phase. The strong bonding zones in the Mg-Ti interfaces provided a high hardening effect, and the strain-hardening rate subsequently increased with an increase in strain.

The work-hardening rate (*n*) of the rolled sheets was also investigated according to the Hollomon relationship, as shown in Equation (1):*σ* =*k*·ε*^n^*
(1)
where *σ* denotes the true stress, ε is the true strain, *k* is the coefficient, and *n* is the work-hardening rate. Thus, *n* can be calculated via a differential calculation method for Equation (1). The results are displayed in Figure 11. It can be seen that *n* was lower than 0.1 for the samples of the as-rolled Mg-(Al-)Ti sheets, and those of the sheets before heat treatment were only about 0.02–0.04, as shown in Figure 11a. The Mg-Ti (1mm Ti) sheet showed a slightly higher work-hardening rate of 0.12. At the same time, the Mg-(Al-)Ti sheets containing thinner Al and Ti (0.01 mm), and their work-hardening rates were significantly larger than those of the sheets containing thick Al or Ti sheets, i.e., about 0.21 before heat treatment and 0.15 after heat treatment. This indicates that the Mg-(Al-)Ti sheets containing thinner Ti/Al sheets or layers (0.01 mm) showed excellent forming performance.

## 4. Conclusions

Dissimilar laminated Mg-(Al-)Ti alloys were successfully prepared by a differential temperature hot-rolling process, and the preheating temperature of Ti was higher 25–100 °C than Mg alloy. The effective interface bonding between Mg/Ti and Al was subsequently achieved when the preheating treatment of Ti alloy was 50 °C higher than that of Mg alloy before rolling at 175 °C.

Well-bonded Mg-Al and Ti-Al interfaces were formed with a thickness of approximately 2–7 μm and 3–5 μm under different rolling processes, respectively, and even formed Mg-Ti interfaces, which was the Mg(Al) and Ti(Al) solid-solution zone, with a thickness of about 5 μm. No intermetallic compounds were generated during the rolling process and the annealing treatment. The bonding interfaces showed 16–30% higher hardness than that of the Ti and Mg matrix.

The Mg-(Al-)Ti rolled sheets showed outstanding mechanical properties. The compressive mechanical properties of the rolled sheets were still better than those of AZ31 and other Mg-based LMCs. The UTS and YTS were about 223–460 MPa and 303–442 MPa, respectively, with an elongation of 0.04–0.17 and high elastic modulus of 52–68 Gpa. The elongation was significantly enhanced when adding an Al interface layer and conduction annealing treatment. The Al layer significantly enhanced the elongation of the Mg-Al-Ti sheets when increasing the thickness from 0.01 mm to 0.05 mm, and the annealing treatment further promoted the elongation at least 80%.

Mg-Ti sheets and Mg-(Al-)Ti with a thin Al layer show an excellent work-hardening rate (*n*) than that of other rolled sheets with a thick Al layer due to the high bonding interfaces and strengthening effect of the Ti or Al pieces around the interfaces.

The differential temperature hot-rolling process provides a simple method to achieve the bonding of dissimilar alloys; however, owing to the unstable thermodynamics Mg-Ti alloy, the rolled Mg-Ti sheets show severe anisotropy with the rolling direction (RD) and normal direction (ND), which restrict its wide application. Nevertheless, the following two issues need to be further studied: the interfacial structure between Mg/Ti and Al, and even between Mg and Ti, which is vital to reveal the bonding mechanism of Mg and Ti. Additionally, the effect of the thickness annealing treatment regime of interdiffusion layers on the mechanical properties needs to be studied in detail.

## Figures and Tables

**Figure 1 materials-15-02805-f001:**
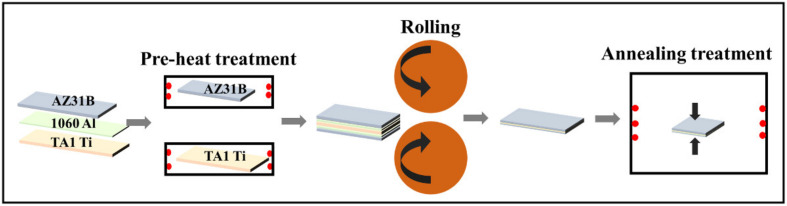
Schematic of differential temperature rolling process of Mg-(Al-)Ti laminate.

**Figure 2 materials-15-02805-f002:**
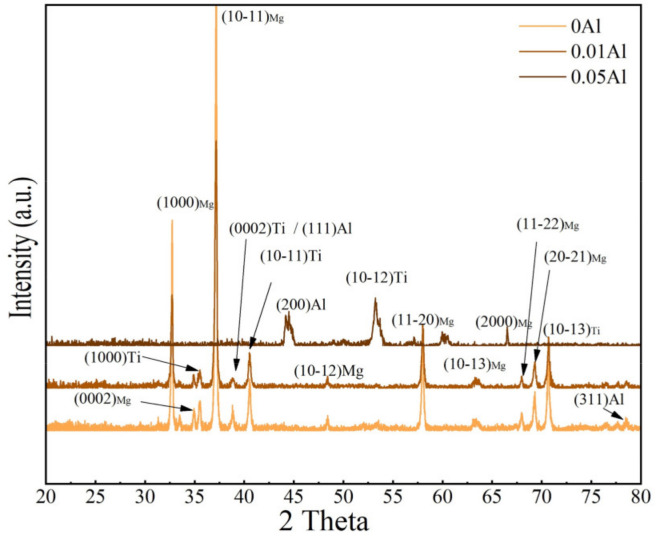
XRD pattern of rolled Mg-(Al-)Ti sheets with different thicknesses of Al layer, and corresponding standard powder diffraction file (PDF) card peaks of α-Ti, Al, and Mg are also marked at the bottom.

**Figure 3 materials-15-02805-f003:**
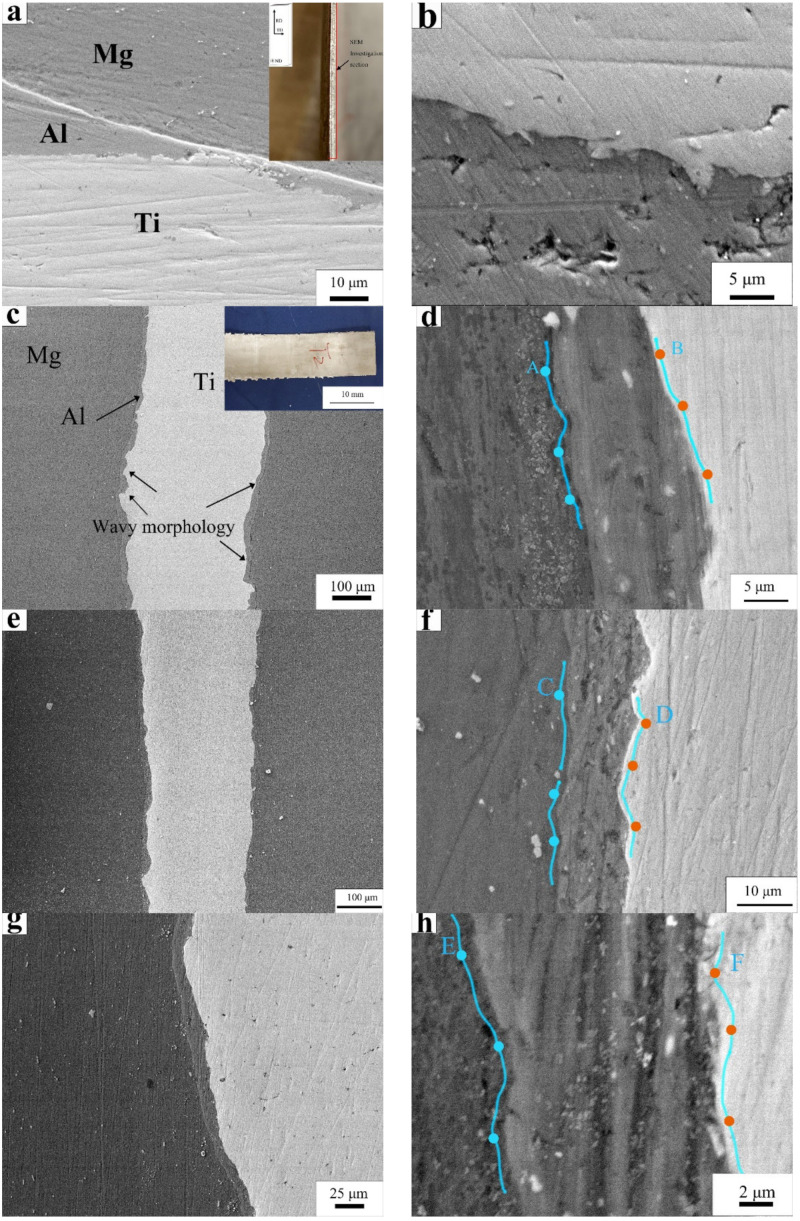
Microstructure and EDS line scanning of Mg-(Al-)Ti sheet rolled at various temperatures: (**a**,**b**) room temperature (RT); inset shows SEM investigation location, section of (ND-TD) plane, ND, TD and RD, meaning normal direction, transversal direction, and rolling direction (**c**,**d**) at 200 °C, (**e**,**f**) 250 °C, and (**g**,**h**) 300 °C. EDS points are indicated by A–F (**d**,**e**,**h**) at a distance of about 1 μm from the interface, the blue line was the EDS scanning location, and the blue dots and orange dots means the EDS point location.

**Figure 4 materials-15-02805-f004:**
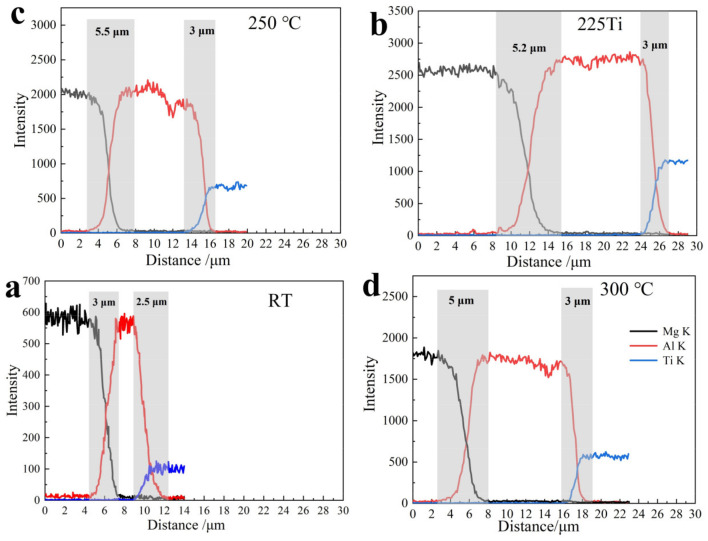
EDS line scanning of Mg-(Al-)Ti sheet rolled with different heat-treated Ti: (**a**) RT; (**b**) 200 °C; (**c**) 225 °C; (**d**) 300 °C.

**Figure 5 materials-15-02805-f005:**
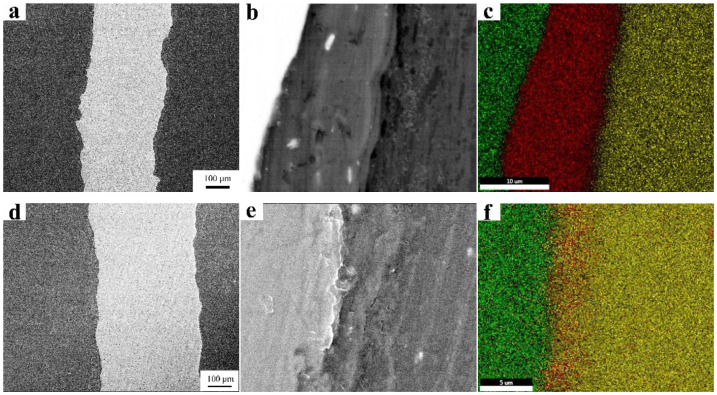
Microstructure and EDS scanning results of the interfaces for Mg-(Al-)Ti sheets with different Al layer thicknesses: (**a**–**c**) 0.05 mm Al layer, (**d**–**f**) 0.01 mm Al layer, and (**g**–**i**) Al layer without interdiffusion. (**j**-**l**) EDS line scanning of Mg-(Al-)Ti sheet rolled with different thicknesses of Al layer: (**j**) 0.05 mm, (**k**) 0.01 mm, and (**l**) for 0 mm.

**Figure 6 materials-15-02805-f006:**
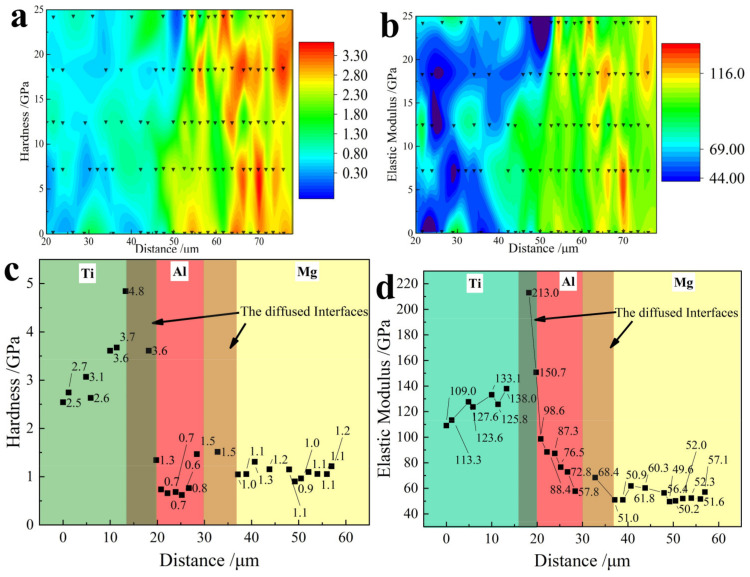
Indentation matrix of Mg-(Al-)Ti interfaces: (**a**) mapping of hardness; (**b**) mapping of elastic modulus, where the triangle denotes the related impresses; (**c**) hardness around the interfaces; (**d**) elastic modulus around the interfaces.

**Figure 7 materials-15-02805-f007:**
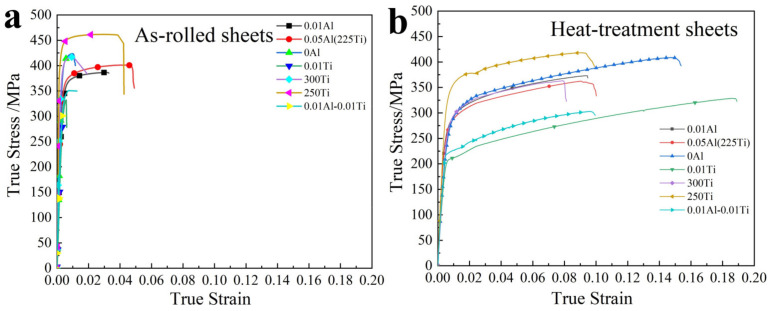
True stress–strain curves of (**a**) as-rolled sheets and (**b**) heat-treated sheets.

**Figure 8 materials-15-02805-f008:**
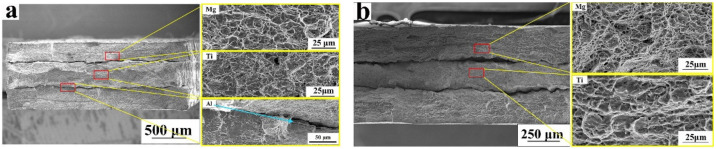
Fracture morphology of the rolled Mg-(Al-)Ti sheets after heat treatment: (**a**) Mg-Al-Ti sheet; (**b**) Mg-Ti sheet, the red boxes were the related Mg, Ti and Al layer.

**Figure 9 materials-15-02805-f009:**
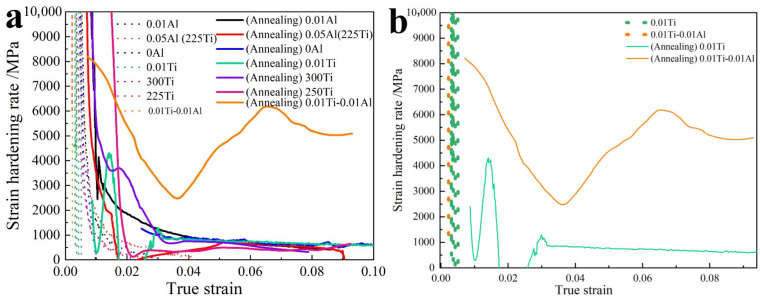
Curves of strain-hardening rate and true strain of the rolled sheets: (**a**) different Mg-(Al-)Ti rolled sheets and their annealing sheets; (**b**) Mg-0.01Ti-0.01Al (0.01Ti-0.01Al) and Mg-0.01Ti- (0.01Ti) rolled sheets.

**Figure 10 materials-15-02805-f010:**
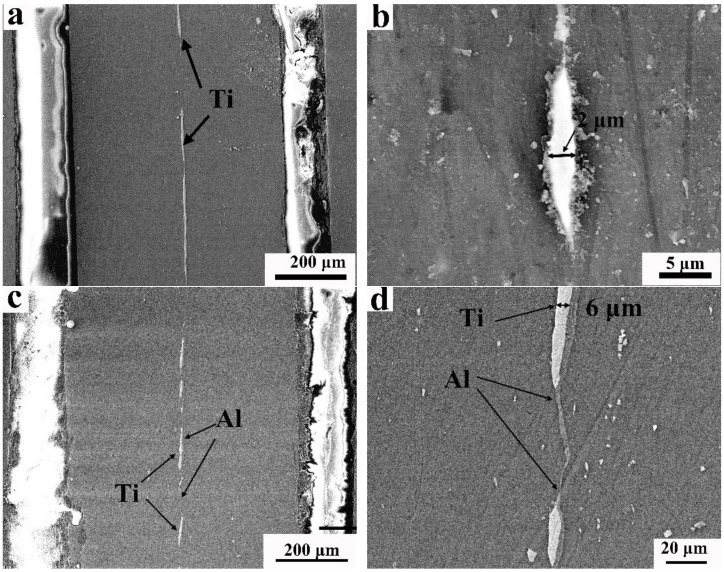
SEM images of cross-section of 0.01Ti and 0.01Ti-0.01Al sheets: (**a**,**b**) SEM image of 0.01Ti sheet; (**c**,**d**) SEM image of 0.01Ti-0.01Al sheet.

**Figure 11 materials-15-02805-f011:**
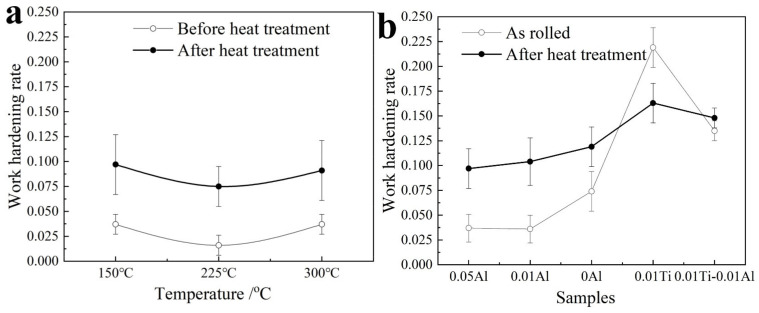
Simulation of true strain and strain to determine the work-hardening rate, (**a**) the samples for different heat treatments of Ti, and (**b**) the sample for different Al layer thicknesses.

**Table 1 materials-15-02805-t001:** Chemical compositions of AZ31B and 1060 Al (wt.%).

Alloy	Al	Si	Ca	Zn	Fe	Be	Mn	Cu	Ti	V	Mg
AZ31B	3.190	0.020	0.040	0.810	0.005	0.100	0.330	0.050	-	-	Bal.
1060 Al	Bal.	0.250	-	0.050	0.350	-	0.030	0.050	0.030	0.050	0.03

**Table 2 materials-15-02805-t002:** Chemical compositions of TA1.

Alloy	Si	O	H	N	C	Fe	Ti
TA1	≤0.001	≤0.001	≤0.001	≤0.001	≤0.001	≤0.001	Bal.

**Table 3 materials-15-02805-t003:** Samples of rolling process.

No.	Samples	The Stacking Layers	Mg/at%	Preheating Temperature
1	0.2Al	Mg-0.2 mmAl-1 mmTi-0.2 mmAl-Mg	53.9	-
2	225Ti	Mg-0.05 mmAl-1 mmTi-0.05 mmAl-Mg	61.5	225 °C for Ti, 200 °C for Mg
3	250Ti	Mg-0.05 mmAl-1 mmTi-0.05 mmAl-Mg	61.5	250 °C for Ti, 200 °C for Mg
4	300Ti	Mg-0.05 mmAl-1 mmTi-0.05 mmAl-Mg	61.5	300 °C for Ti, 200 °C for Mg
5	0.05Al	Mg-0.05 mmAl-1 mmTi-0.05 mmAl-Mg	61.5	250 °C for Ti, 200 °C for Mg
6	0.01Al	Mg-0.01 mmAl-1 mmTi-0.01 mmAl-Mg	63.3	250 °C for Ti, 200 °C for Mg
7	0Al	Mg-1 mmTi-Mg	63.7	250 °C for Ti, 200 ^o^C for Mg
8	0.01Ti	Mg-0.01 mmTi-Mg	95.5	250 °C for Ti, 200 °C for Mg
9	0.01Ti-0.01Al	Mg-0.01 mmAl-0.01 mmTi-Mg	95.0	250 °C for Ti, 200 °C for Mg

**Table 4 materials-15-02805-t004:** Point EDS results around the interfaces (at.%).

Points	Mg	Al	Ti
A	96.2 (±2)	2.9 (±1)	0.9 (±0.6)
B	3.7 (±1)	5.8 (±1)	90.5 (±2)
C	91.4 (±2)	8 (±1)	0.5 (±0.3)
D	2.4 (±1)	2.4 (±1)	95.2 (±2)
E	95.8 (±2)	3 (±1)	1.2 (±0.5)
F	2.9 (±1)	2.7 (±1)	94.4 (±1)

**Table 5 materials-15-02805-t005:** Mechanical properties of Mg-(Al-)Ti rolling sheets before heat treatment.

Samples	UTS/MPa	YTS/MPa	Elongation to Break
0.05Al	399	358	0.05
0.01Al	387	354	0.03
0Al	424	408	0.01
0.01Ti	331	329	0.06
0.01Ti-0.01Al	349	342	0.01
225Ti	399	358	0.05
250Ti	460	442	0.04
300Ti	424	401	0.01

**Table 6 materials-15-02805-t006:** Mechanical properties of Mg-(Al-)Ti rolled sheets after heat treatment and other reported Mg-based composite rolled sheets.

Samples	Mg Content/at.%	E/GPa	UTS/MPa	YTS/MPa	Elongation to Break	Ref.
0.05Al	61.5	68	363	267	0.09	In this work
0.01Al	63.3	62	373	271	0.09	
0Al	63.7	56	408	275	0.15	
0.01Ti	95.5	50	328	288	0.18	
0.01Ti-0.01Al	95.0	52	303	223	0.1	
225Ti	61.5	68	363	267	0.09	
250Ti	61.5	67	418	339	0.09	
300Ti	61.5	65	363	288	0.08	
AZ31B	96	~44 ^◆^	322	180	0.17	[34]
Mg-Steel	~60 *	--	265	260	0.048	[17]
Mg-Al-TC4	~22 *	--	459	380	0.08	[27]
TA2/5052Al/AZ31/5052Al/TA2	~58	--	429	<200	0.37	[16]

* strength obtained from the engineered stress–strain curves; ^◆^ estimated value.

## Data Availability

The data used to support the findings of this study are included within the article.

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
