# Peer review of "Effect of Al Layer Thickness on the Bonding and Mechanical Behavior of a Mg-(Al-)Ti Laminated Sheet Prepared by Hot-Rolling after Differential Preheating Treatment"

_materials, 2022, doi:10.3390/ma15082805_

Round 1
Reviewer 1 Report
- Add more current references (2020 – 2021).
- Highlight better which is the novelty of the work?
- What is the status of the literature according to your work? Make a comparison between the results obtained by you and another previous research.
- In which applications do you think the studied materials are suitable?
- Introduce into a table all the results obtained from Nanoindentation tests.
- Complete the conclusions with the limitations of the proposed methodology. Also write future research.
- Add Miller indices on Figure 2 XRD pattern.
- Try increasing the resolution in Figure 2.
- Generally, the quality of the writing could be improved.
Author Response
Dear reviewer,
Thank you for your reviewing the paper and comments. According to these comments, we have revised the related places and made the marks. At the same time, some writing and grammar problems have been revised and marked, the detailed comments and respond were stated as follows.
The revising was approved by all co-authors. Please let me know that if there is any questions about the revision and the submission.
Kind regards.
Sincerely yours,
Zhiyong Xue
Comments and responds
- Add more current references (2020 – 2021).
Thank you for your suggestion, we have added 6 references in the paper, and the related places have been revised.
- Highlight better which is the novelty of the work?
The highlights were stated as follows:
- The simple method to achieve the dissimilar metals bonding of Mg alloy and high melt point Ti alloy with differential temperature hot-rolling process, its width of interfaces was controlled to about 2-7 μm for Mg-Al and 3-5 μm Al-Ti interfaces, respectively.
- The rolled Mg-(Al-)Ti shows higher 30-88% strength than AZ31B alloy, and greater 13-55% elastic modulus than AZ31B alloy, with no decreasing of elongation condition.
- What is the status of the literature according to your work? Make a comparison between the results obtained by you and another previous research.
The comparison explanation was added, in Page 10.
- In which applications do you think the studied materials are suitable?
The Mg-(Al)-Ti sheets could apply to the field of aerospace, like the space exploration payloads platforms, and the drone wing skin, which the application has a vital consideration for high elastic modulus or stiffness.
- Introduce into a table all the results obtained from Nanoindentation tests.
Thank you for your suggestion, we have listed all the effective nanoindentation results in a single word, which contains two Table, Table S1 was the original data for the matrix analysis, and Table S2 was the original data for line scanning.
- Complete the conclusions with the limitations of the proposed methodology. Also write future research.
In the conclusions section, we added a paragraph to state the limitations of the proposed methodology and the future research.
- Add Miller indices on Figure 2 XRD pattern.
The related Miller planes indices have been added, in Page 4.
- Try increasing the resolution in Figure 2.
The resolution has been enhanced, in Page 4.
- Generally, the quality of the writing could be improved.
We checked and revised the whole manuscript for writing, including grammar, sentence and word. The related places has revised and marked.

Reviewer 2 Report
Authors reported their research work on “Effect of Al layer thickness on the bonding and the mechanical behaviors of Mg-(Al-)Ti laminated sheet prepared by hot-rolling after the differential preheating treatment”. The work is well organized with compact results and discussion section. But few things need to be addressed: 1. Degree (º) symbols should be corrected throughout the manuscript. 2. Authors should explain in the introduction section about the novelty of this work. 3. In Fig. 1: the embedded marking in the figure is not readable. Authors should change that. 4. What is the reason behind selecting temperature regime” specially, 20ºC? 5. Did authors try higher temperature than 300/400ºC? 6. From the EDS line scan, authors showed diffusion layer thickness. But authors didn’t mention anything about IMC’s. How is the IMC fraction and what kind of IMC’s formed in Al-Mg and Al-Ti side? 7. Legends in Fig.4, 5, 6 and 7 are not readable. Authors should enlarge the size for better readability. 8. In Fig.3: SEM microstructure have been shown. It would be great if authors exhibit a macro cross-section of the rolled samples and marked where SEM has been performed? 9. In Page no.2, line no.6, TA1 should be replaced with Ti for better understanding. 10. In Fig.4, interestingly diffusion thickness does not change much from 20ºC to 300ºC. Authors should provide an explanation behind that. 11. Fig. 5 (f) and (i) is not clear. 12. In Fig.6, how many nano-indentations have been taken for each case should mention? 13. Hardness mapping in Fig. 6(b) and (c) is bit confusing. Also, the color marker with the hardness mapping is not readable. 14. Why there is so much noise in Fig. 9(b)? Authors should explain. 15. Conclusion should be more concise for better understanding.Author Response
Dear reviewer,
Thank you for your reviewing the paper and comments. According to these comments, we have revised the related places and made the marks. At the same time, some writing and grammar problems have been revised and marked, the detailed comments and respond were stated as follows.
The revising was approved by all co-authors. Please let me know that if there is any questions about the revision and the submission.
Kind regards.
Sincerely yours,
Zhiyong Xue
Authors reported their research work on “Effect of Al layer thickness on the bonding and the mechanical behaviors of Mg-(Al-)Ti laminated sheet prepared by hot-rolling after the differential preheating treatment”. The work is well organized with compact results and discussion section. But few things need to be addressed:
- Degree (º) symbols should be corrected throughout the manuscript.
Thank you for your suggestion, according to this comment, the symbols: oC and o, have been revised throughout the manuscript,
- Authors should explain in the introduction section about the novelty of this work.
We rewrite the last paragraph in introduction section, basing the comment.
- In Fig. 1: the embedded marking in the figure is not readable. Authors should change that.
Fig.1 has been revised, in Page 2.
- What is the reason behind selecting temperature regime” specially, 20ºC?
The temperature regime was selected basing our previous study work on the Mg and Mg-Al-Ti rolling, we tested room temperature (RT) - 450 oC, and finally, we chosen the temperature zone: RT - 300 oC, the higher temperature leads to some problems, like higher surface oxidation of the raw materials, and even forming the intermetallic compounds around the interfaces.
20ºC was just denotes room temperature. We canceled it and just kept “RT” in the whole manuscript.
- Did authors try higher temperature than 300/400ºC?
Actually, we try the temperature 450 oC for Ti, while the surface of Ti has oxidated with a dark skin, and the rolling process also subsequently conducted, the rolled Mg-Al-Ti sheets occurred fracture cross the interface direction with conducting the mechanical test. Nevertheless, we will continue to study the effect of high temperature on the bonding Mg-Al-Ti sheets in later, in detail.
- From the EDS line scan, authors showed diffusion layer thickness. But authors didn’t mention anything about IMC’s. How is the IMC fraction and what kind of IMC’s formed in Al-Mg and Al-Ti side?
The EDS line scanning shows the interdiffusion of Mg-Al, Ti-Al, or Ti-Mg, which were observed in most of our samples. While no compounds were observed. The rolling temperature was conducted at 175 oC, it is low to form the IMCs, actually, according the references [*], the IMC could form with a high temperature of 450 oC.
In our study, basing the analysis SEM/EDS and XRD around the interfaces zone, it is the solid solution zone due to the atom diffusing, the EDS results also were showed in Table 4.
(*Nie, H.; Zheng, L.; Kang, X.; Hao, X.; Li, X.; Liang, W. In-Situ Investigation of Deformation Behavior and Fracture Forms of Ti/Al/Mg/Al/Ti Laminates. T. Nonferr. Metal. Soc. 2021, 31 (6), 1656-1664.)
- Legends in Fig.4, 5, 6 and 7 are not readable. Authors should enlarge the size for better readability.
The related figures have revised, including the figure legends and texts in the figures.
- In Fig.3: SEM microstructure have been shown. It would be great if authors exhibit a macro cross-section of the rolled samples and marked where SEM has been performed?
Thank you for your suggestion, we provided a insert on the Figure 3a, for showing the samples location.
- In Page no.2, line no.6, TA1 should be replaced with Ti for better understanding.
Thank you for your suggestion, it has been revised in the related places.
- In Fig.4, interestingly diffusion thickness does not change much from 20ºC to 300ºC. Authors should provide an explanation behind that.
Adding the following interpretation: Generally, the diffusion interface’s width has little changed with increasing of pre-heat treatment, it is because that the temperature, including rolling temperature has little effect on the atom inter-diffusion, which the diffusion occurred with a short time at the rolling process. It could be verified in other Mg-Al-Ti rolling sheets, for example, the Ti/Al/Mg laminated composites, which the inter-diffusion thicknesses, including Mg-Al and Al-Ti interfaces, were rarely changed at even higher temperature, 400 oC [25].
- Fig. 5 (f) and (i) is not clear.
Fig.5 (f) and (i) were obtained from larger magnification than Fig.5(c) for investigating the Al atom diffusion, due to decreasing the thin Al layer, their resolutions were relatively low, nevertheless, we found the original images, and enhancing them with adjusting the bright and contrast.
- In Fig.6, how many nano-indentations have been taken for each case should mention?
Thank you for your suggestion, The figure includes about 120 points, and we revised the whole images for better express, and Figure 6a and b shows the whole points with little dark triangles.
- Hardness mapping in Fig. 6(b) and (c) is bit confusing. Also, the color marker with the hardness mapping is not readable.
Thank you for your suggestion, we revised the related images.
- Why there is so much noise in Fig. 9(b)? Authors should explain.
We remade a new images, Figure 9b, at the same time, we also added an new series microstructure images for better expressing the strain hardening rate of 0.01Ti and 0.01Ti-0.01Al samples.
- Conclusion should be more concise for better understanding.
We rewrite the conclusion section more concise.
Reviewer 3 Report
The aim of the paper, i.e. the manufacturing of Mg-(Al-)Ti laminated sheets with great bonding interfaces were prepared by a simple differential temperature hot-rolling process, is quite interesting. The authors used the Al layer of different thickness to study their effect on bonding between Mg and Ti sheets. Additionally, they applied the hot rolling with differential preheating of raw materials before hot rolling, and then the authors successfully prepared the rolled Mg-(Al-)Ti sheets. The bonding interfaces and mechanical properties were also systematically studied.
The abstract summarize the work. The purpose of the study is clearly outlined and the findings of prior work are well discussed.
There are no errors in logic or experimental procedure. The authors accurately explain how the data were collected. There is sufficient information that the experiment can be reproduced.
All topics are very well presented. Unfortunately, the obtained results are rather poorly discussed with reference to the literature. This part of the paper must be improved.
The summary and conclusions are sound and justified. All presented figures are very good quality and they prove their point.
The paper is written in good English. The manuscript is easily readable concerning language, style and presentation. The references are appropriate and up to date.
Following are some of my comments.
- Lines 40, 42. Please use subscripts in stoichiometric formulas.
- Lines 88, 94, 139, 174, 176, 179, 253, 255, 259, 272, . Please use superscripts to denote Celsius degrees.
- The obtained results are rather poorly discussed with reference to the literature. This part of the paper must be improved.
Author Response
Dear reviewer,
Thank you for your reviewing the paper and comments. According to these comments, we have revised the related places and made the marks. At the same time, some writing and grammar problems have been revised and marked, the detailed comments and respond were stated as follows.
The revising was approved by all co-authors. Please let me know that if there is any questions about the revision and the submission.
Kind regards.
Sincerely yours,
Zhiyong Xue
- Lines 40, 42. Please use subscripts in stoichiometric formulas.
Thank you for your reviewing and suggestion, we have corrected the nonstandard subscripts, in the introduction section.
- Lines 88, 94, 139, 174, 176, 179, 253, 255, 259, 272, . Please use superscripts to denote Celsius degrees.
We have revised the above the related problems.
- The obtained results are rather poorly discussed with reference to the literature. This part of the paper must be improved.
It is a great suggestion, we added the comparison statement in the Page 10, line 297-303.

Round 2
Reviewer 1 Report
The presented data are original and interesting. The manuscript has been significantly improved and is suitable for publication in the present Journal.
Reviewer 2 Report
Authors provide a significant amount of changes in the manuscript and properly address all the reviewer's comments. It should be accepted in its present format.